# MoS$_2$ nanopore identifies single amino acids with sub-1 Dalton resolution

**Fushi Wang[1], Chunxiao Zhao[1], Pinlong Zhao[1], Fanfan Chen[1], Dan Qiao** [1] **& Jiandong Feng** [1,2] ✉

The sequencing of single protein molecules using nanopores is faced with a huge challenge due to the lack of resolution needed to resolve single amino acids. Here we report the direct experimental identification of single amino acids in nanopores. With atomically engineered regions of sensitivity comparable to the size of single amino acids, MoS$_2$ nanopores provide a sub-1 Dalton resolution for discriminating the chemical group difference of single amino acids, including recognizing the amino acid isomers. This ultra-confined nanopore system is further used to detect the phosphorylation of individual amino acids, demonstrating its capability for reading post-translational modifications. Our study suggests that a sub-nanometer engineered pore has the potential to be applied in future chemical recognition and de novo protein sequencing at the single-molecule level.

Due to the lack of techniques that fully account for the complexity of proteomes, proteomics has yet to reach the power of genomics and transcriptomics. The development of protein sequencing technologies can provide transformative information for proteomics that may revolutionize biological research and precision medicine applications[1]. Edman degradation and mass spectrometry[2,3], as current mainstream approaches for protein sequencing, have deficiencies in their detection speed, read length or achieving routine, complete proteome quantification at low abundance. To overcome these issues, several disruptive single-molecule approaches[4–9] have been proposed to potentially sequence and identify individual proteins. A strong impetus for extending the successful nanopore DNA sequencing to potential protein sequencing stands on the long-read length and the portability[1].

Although the detection of proteins[10–13] and short peptides[14] has already been realized in nanopores, compared with DNA sequencing, nanopore sequencing of proteins remains elusive mainly due to the two extra challenges faced for the read of permutations of 20 amino acids instead of four nucleobases and the translocation control of heterogeneously charged peptides. Similar to the enzyme-based stepping control of DNA translocation, molecular motors, such as ClpX[15] and proteasome[16], were employed to unfold and pull proteins through nanopores. Peptides linked to DNA could also be

pulled through nanopores by DNA helicase[17] or polymerase[18], which enabled the discrimination of single-amino acid substitutions. This approach offers an effective route to meet the temporal resolution requirement for nanopore protein sequencing, although the use of the enzyme limits the reading speed. In terms of the spatial resolution, many efforts have been made to improve the sensitivity of nanopore for reading proteins, achieving the discrimination of amino acid substitutions in a carrier polymer[19], different sizes of short uniformly charged homopeptides[20], and post-translational modifications[21]. However, for directly resolving the tiny differences among the 20 natural amino acids, the spatial resolution remains the main bottleneck that restricts the development of nanopore protein sequencing. For potential de novo protein sequencing using a similar approach to DNA sequencing[22], as noted by Brinkerhoff et al., a MspA nanopore simultaneously measures ~8 amino acids within its region of sensitivity, and the number of required signals to resolve single amino acid will be around eight power of twenty that is impractically large[17]. However, if the region of sensitivity of the nanopore can be shortened to the size level of a single amino acid, then the task will be greatly simplified to identifying only 20 amino acids. This calls for developing nanopores with single amino acid region of sensitivity, including sub-nanometer length and molecular scale orifice.

[1]Laboratory of Experimental Physical Biology, Department of Chemistry, Zhejiang University, 310027 Hangzhou, China. [2]Research Center for Quantum Sensing, Research Institute of Intelligent Sensing, Zhejiang Lab, 311121 Hangzhou, China. ✉e-mail: jiandong.feng@zju.edu.cn

Unlike biological nanopores made by the assembly of proteins in which the region of sensitivity is still limited by the constituting amino acids, atomically constructed synthetic nanopores, such as molybdenum disulfide ($MoS_2$) nanopores[23,24], offer a direct solution toward meeting these goals. Here we designed a $MoS_2$ nanopore system in which the dimension of the pore is comparable to that of single amino acids. This pore architecture empowered the direct identification of single amino acids in nanopores.

## Results

### Identifying single amino acids in MoS₂ nanopores

Nanopore experiments were performed for translocating amino acids in a typical electrophoretically driven configuration, as shown in Fig. 1a. Information from relative current blockade ($\Delta I/I_0$) and dwell time ($\Delta t$) are used for characterizations, and the error for $\Delta I/I_0$ is calculated by the standard deviation (Fig. 1b–g, Supplementary Figs. 1 and 2). Initially, we translocated homopeptides of different lengths and found for glycine (G), Gly-Gly (GG), and Gly-Gly-Gly (GGG), the induced current blockades were $0.129 \pm 0.021$ nA, $0.127 \pm 0.016$ nA, and $0.127 \pm 0.021$ nA, respectively (Fig. 1d). The comparable values of the

current blockade from homopeptides with different lengths imply that for the current $MoS_2$ nanopore, the region of sensitivity is equal to or shorter than the size of a single amino acid (Supplementary Fig. 3), which is in sharp contrast to the length-dependent blockade with a biological pore[20]. The ultrathin region of sensitivity observed in our experiments is comparable with the $MoS_2$ nanopore sensitivity analyzed in molecular dynamics simulations[25,26] (Supplementary Note 2). Further experiments under different voltages were performed to confirm that the observed events indeed came from single amino acid translocations (Supplementary Figs. 4 and 5), as the amplitudes of the current blockade ($\Delta I$) increased with the increase of the potential from 200 to 300 mV.

Considering the heterogeneity of $MoS_2$ nanopore devices, we carried out a series of amino acid identification experiments in 41 different $MoS_2$ nanopores (Supplementary Figs. 6–10) to ensure the reproducibility of our experimental system. The effective diameters of nanopores were controlled to range from sub-nanometer to 1.6 nm, and we found that the appropriate size (Supplementary Figs. 8–10) is critical to the sensitivity of the pore (Supplementary Figs. 11–13). For a nanopore size comparable to

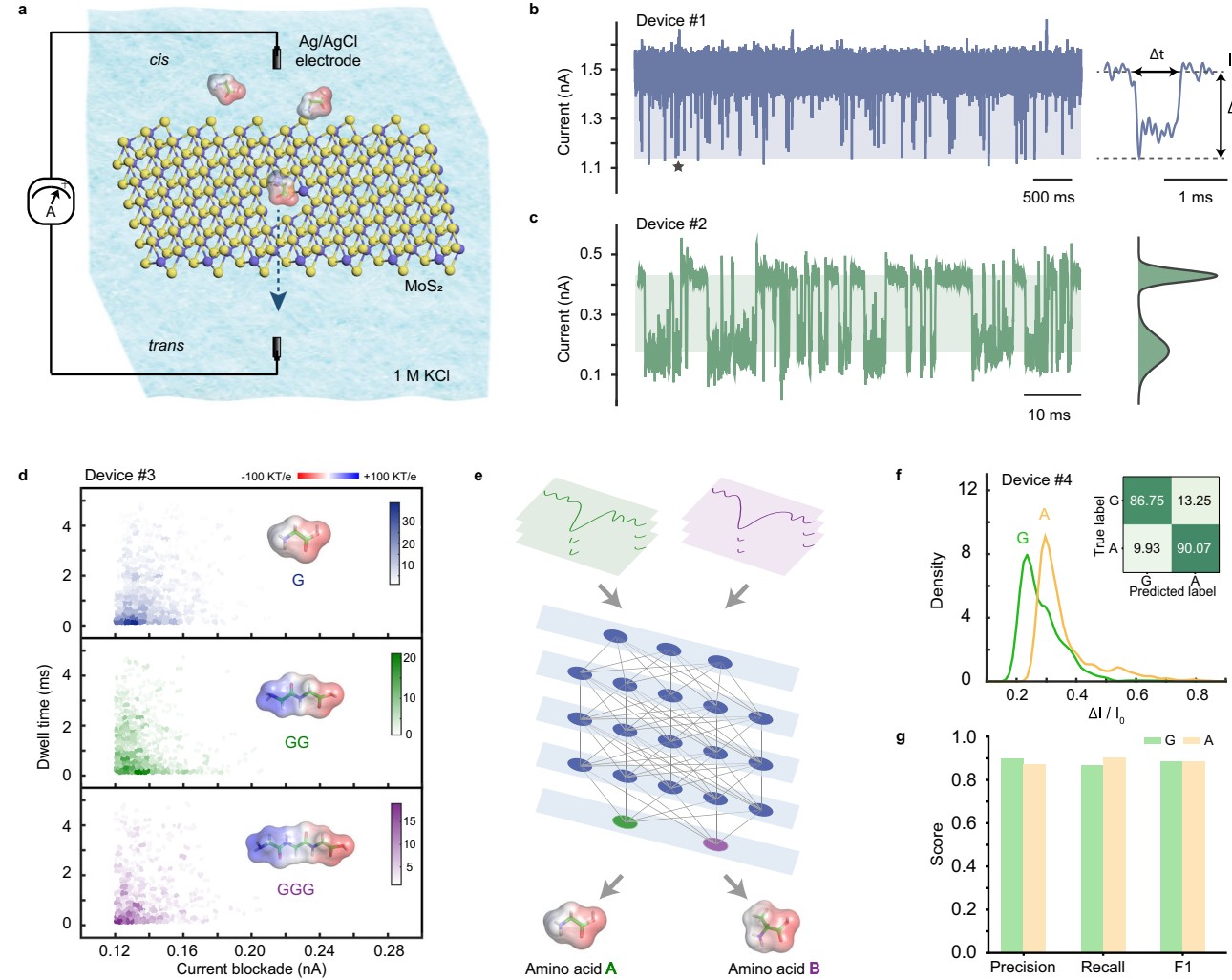

**Fig. 1 | Detection of single amino acids in MoS₂ nanopores. a** Schematic of the experimental setup (not to scale). Amino acids are electrophoretically driven through a $MoS_2$ nanopore. **b** An example trace recorded with the addition of 2 μM E in a -1.1 nm nanopore (Device #1) and a typical current blockade selected from the trace (marked by pentagram). **c** An example trace recorded with the addition of 2 μM A in a -0.5 nm nanopore (Device #2) with the current histogram. The bimodal distribution is from the baseline current and the blockade current. **d** Heatmap of dwell time versus current blockade of G, GG, and GGG, respectively (Device #3). Mean peak values: G, $0.129 \pm 0.021$ nA; GG, $0.127 \pm 0.016$ nA; GGG, $0.127 \pm 0.021$ nA. **e** A flow diagram of amino acids identification using SAAINet. **f** Histograms of $\Delta I/I_0$ obtained from nanopore experiments of G and A (Device #4) displayed as their fitting curves. Mean peak values: G, $0.229 \pm 0.016$; A, $0.295 \pm 0.021$. The average identification accuracy is 88.41%. **g** Graphical summary of precision, recall, and F1 score from SAAINet for the data in (**f**).

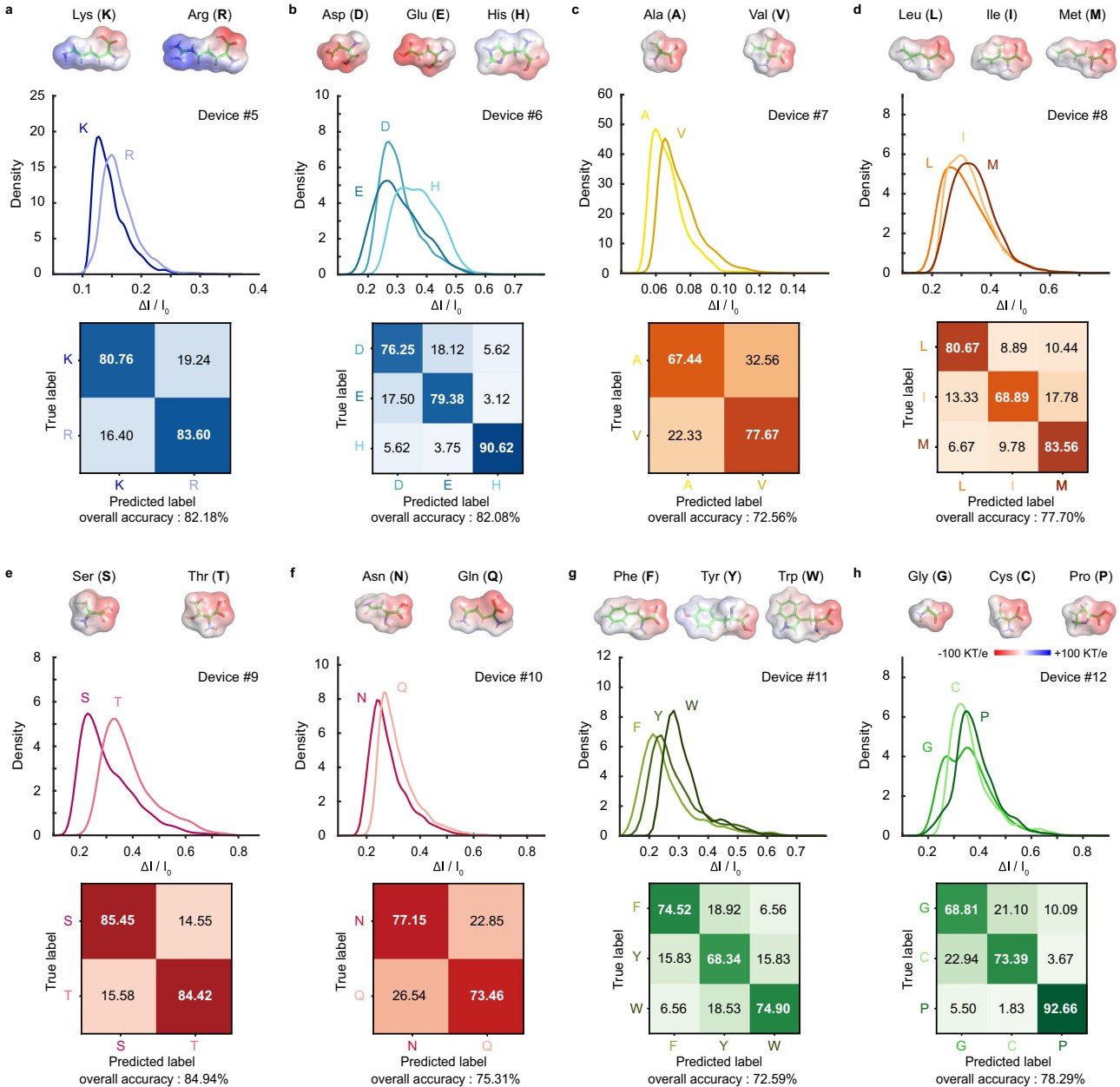

**Fig. 2 | Identification of single amino acids. a–h** Histograms of $\Delta I/I_0$ obtained from nanopore experiments performed for 20 amino acids, and the results are displayed in the following groups: electrically charged amino acids (**a**, **b**), hydrophobic nonaromatic amino acids (**c**, **d**), polar uncharged amino acids (**e**, **f**), hydrophobic aromatic amino acids (**g**), and special group consists of G, C, and P (**h**). The distributions of histograms were fitted into curves. Mean peak values are as follows, K: $0.127 \pm 0.028$, R: $0.154 \pm 0.023$ (Device #5); D: $0.273 \pm 0.035$, E: $0.252 \pm 0.042$; H: $0.298 \pm 0.026$ and $0.389 \pm 0.068$ (Device #6); A: $0.058 \pm 0.009$, V: $0.065 \pm 0.009$ (Device #7); L: $0.230 \pm 0.076$, I: $0.284 \pm 0.065$, M: $0.329 \pm 0.069$ (Device #8); S: $0.228 \pm 0.033$, T: $0.327 \pm 0.048$ (Device #9); N: $0.241 \pm 0.031$, Q: $0.268 \pm 0.024$ (Device #10); F: $0.212 \pm 0.052$, Y: $0.241 \pm 0.054$, W: $0.284 \pm 0.047$ (Device #11); G: $0.257 \pm 0.019$ and $0.356 \pm 0.077$, C: $0.326 \pm 0.053$, P: $0.349 \pm 0.060$ (Device #12). Confusion matrixes are attached to each histogram. The results of K and R were acquired under $-200$ mV applied to the trans compartment due to their positive charges at pH 7.8, and that of the remaining 18 amino acids were obtained under $+200$ mV due to their negative charges at pH 7.8.

the amino acids being detected, the resulting histogram of the current trace is bimodal (Fig. 1c, Supplementary Fig. 14), and the ionic current can be nearly blocked to 0 nA. The $MoS_2$ nanopores can work continuously for dozens of hours and allow the recording of more than 70,000 events. To facilitate the comparisons and discussions, the 20 natural amino acids are grouped into five groups: electrically charged amino acids (Fig. 2a, b), hydrophobic nonaromatic amino acids (Fig. 2c, d), polar uncharged amino acids (Fig. 2e, f), hydrophobic aromatic amino acids (Fig. 2g), and the other special amino acids (Fig. 2h).

For each group of data, the peaks of histograms of relative current blockades are separated despite the overlaps. We performed a *z*-test on all data which show *P*-values all less than 0.0001, indicating the high statistical significance of the observed difference. To call the specific type of an amino acid from a single event, we introduced a deep learning network, SAAINet (see Supplementary Note 4), to identify the individual events (Fig. 1e, Supplementary Figs. 15–20), as shown in the confusion matrix results attached to each histogram. Different from the conventional machine learning approaches applied to nanopores[24], our SAAINet not only extracts the specified features but

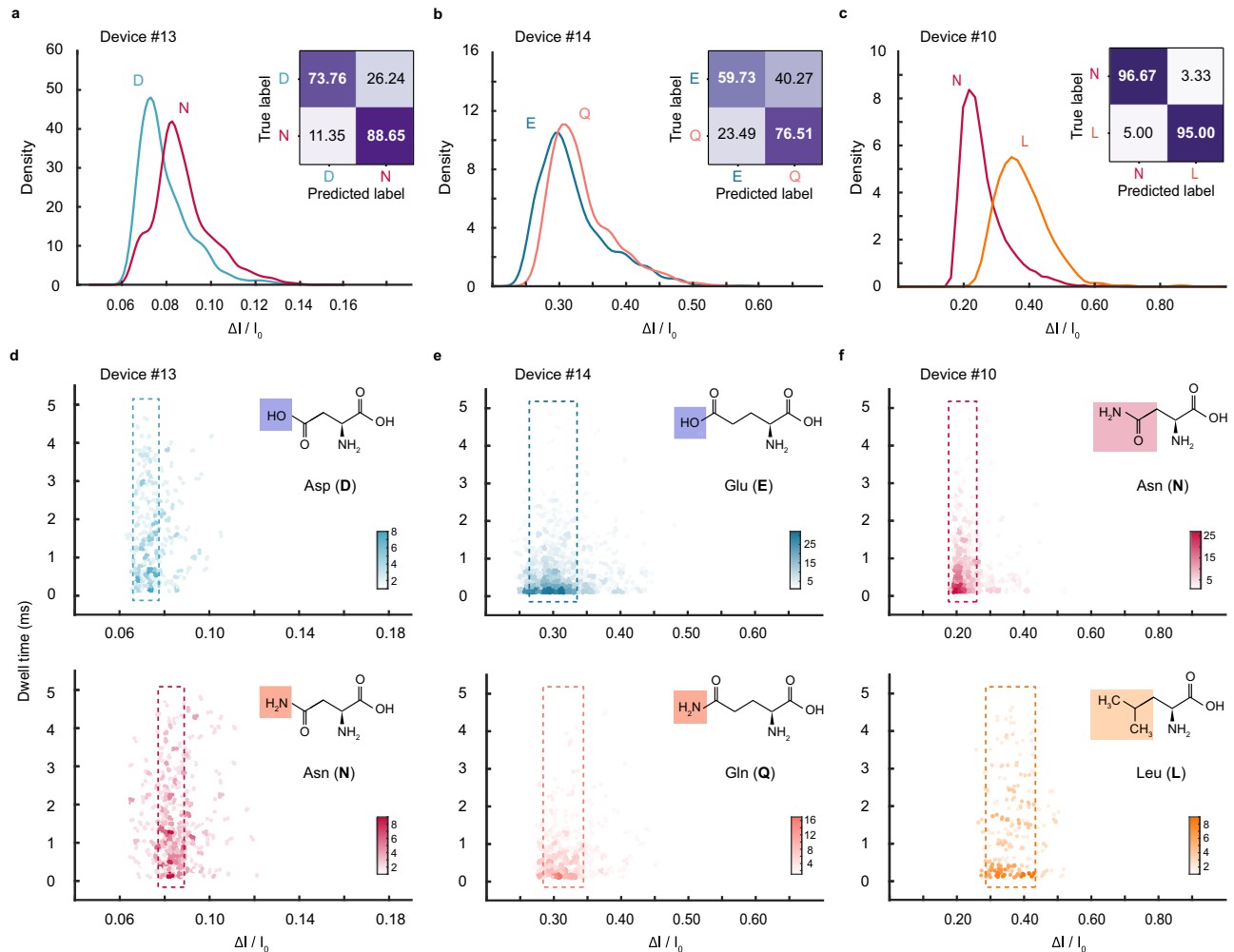

**Fig. 3 | Ultrahigh resolution of MoS₂ nanopores for identifying chemical groups. a–c** Histograms of $\Delta I/I_0$ values for D and N (Device #13), E and Q (Device #14), and N and L (Device #10), respectively. The distributions of histograms were fitted into curves. Mean peak values: D, 0.072 ± 0.006; N, 0.083 ± 0.006; E, 0.300 ± 0.037; Q, 0.314 ± 0.031; N, 0.218 ± 0.042; L, 0.360 ± 0.078. The average identification accuracy is 81.21%, 68.12%, and 95.83%, respectively. **d–f** Heatmaps for D and N (Device #13), E and Q (Device #14), and N and L (Device #10), respectively. The width of the box in figure (**d–f**) represents the standard deviation.

also directly converts the ionic current into vectors, showing a high universality for analysis of nanopore data (Supplementary Figs. 17–20).

The electric charge amino acids carry varies widely. At pH 7.8 (buffer: 1 M KCl, 10 mM Tris-HCl, 1 mM EDTA), aspartic (D, 133.11 Da), glutamic (E, 147.13 Da), and histidine (H, 155.15 Da) are negatively charged, while lysine (K, 146.19 Da) and arginine (R, 174.20 Da) are positively charged. This charge contrary requires inverting the bias direction for electrophoretically driving the related amino acids to translocate through the pore (Supplementary Fig. 21), and thus these five amino acids (D, E, H, K, R) are divided into two subgroups. Readings of these five amino acids have been reproduced in five different MoS₂ nanopore devices (Fig. 2a, b, Supplementary Fig. 22). For K and R, the mean values of $\Delta I/I_0$ are 0.127 ± 0.028 and 0.154 ± 0.023, and the average identification accuracy of K and R reaches 82.18% (Fig. 2a). The peaks of D and E (0.273 ± 0.035 and 0.252 ± 0.042) are too close to be distinguished directly (Fig. 2b). However, the histogram of H shows an obvious difference from the other two amino acids, featuring two adjacent peaks with mean $\Delta I/I_0$ values of 0.298 ± 0.026 and 0.389 ± 0.068, which can be caused by the different pore entering orientations of H (Supplementary Fig. 23). With SAAINet that accounts for more characters of the nanopore signal, D, E, and H can be clearly identified with an overall accuracy of 82.08%.

Among the five amino acids in the hydrophobic nonaromatic group (Fig. 2c, d), methionine (M, 149.21 Da) has a chance to

induce a relatively large current blockade because of the molecular interaction between the sulfur atom of M and the MoS₂ pore edge. In this set of experiments, we also found that the isomeric amino acids leucine (L, 131.18 Da) and isoleucine (I, 131.18 Da) can induce distinguishable current blockades (Fig. 2d). In seven MoS₂ nanopore devices (pore size ranging from 0.5–1.4 nm), we observed that I showed a higher value of $\Delta I/I_0$ than L in five devices (Fig. 2d, Supplementary Fig. 24a), but the results were reversed in the other two devices (Supplementary Fig. 24b). The latter case happened in the nanopores with -0.6 nm diameter. We attribute this change to the possibility that the sizes of nanopores can influence the pore entering orientation[13] of amino acids. With SAAINet, the average identification accuracy of these two isomeric amino acids improves to 87.25% (Supplementary Fig. 24a).

For polar uncharged amino acids, serine (S, 105.09 Da), threonine (T, 119.10 Da), asparagine (N, 132.12 Da), and glutamine (Q, 146.15 Da), the relative current blockades are positively correlated with the volumes of the molecules (Fig. 2e, f, Supplementary Fig. 25). It is worth noting that the difference is only one methyl group for S and T, and one methylene group for N and Q. The ability to identify this group of amino acids indicates that MoS₂ nanopores can recognize single chemical group with molecular weight as low as 14.01 Da (Fig. 2e). Five nanopore devices used for the experiments of S and T showed reproducible

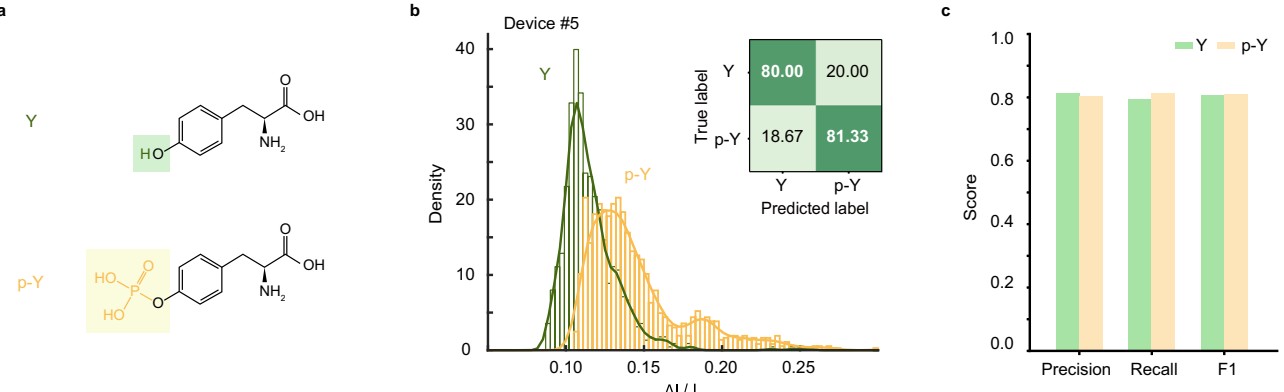

**Fig. 4 | Discrimination of amino acid phosphorylation. a** Chemical structures of Y and p-Y. **b** Histograms of $\Delta I/I_0$ for Y and p-Y (Device #5). Mean peak values: Y, 0.106 ± 0.012; p-Y, 0.128 ± 0.019. The average identification accuracy is 80.69%. **c** Graphical summary of precision, recall, and F1 score from SAAINet.

discriminations of this chemical group difference in the relative current blockades (Supplementary Fig. 25a), and SAAINet brings the overall accuracy to 84.94% (Fig. 2e).

The results for phenylalanine (F, 165.20 Da), tyrosine (Y, 181.19 Da), and tryptophan (W, 204.20 Da) are in line with the expectations based on their volumes (Fig. 2g), W > Y > F in the values of $\Delta I/I_0$, and the overall identification accuracy reaches 72.59%. Note that there is only one hydroxyl group difference between F and Y; however, their $\Delta I/I_0$ still reveals a clear difference (Fig. 2g). This result was reproduced in five devices (Supplementary Figs. 11 and 26) and proved again that MoS₂ nanopores have superior sensitivity for revealing chemical group difference.

Among the remaining three amino acids, glycine (G, 75.07 Da), cysteine (C, 121.16 Da), and proline (P, 115.13 Da), P produced the largest $\Delta I/I_0$ value of 0.349 ± 0.060, due to the five-membered ring in its structure that makes it the largest of the three amino acids in volume (Fig. 2h, Supplementary Fig. 27). Similar with M, C also has a chance of blocking the nanopores due to the contained sulfur atom[27]. Overall, the relative current blockade was found to be a robust feature (Supplementary Note 3, Supplementary Figs. 28 and 29) for identifying single amino acids which increases with the increase of the volume of amino acids. Our conclusion is consistent with the molecular dynamics simulation results obtained by Barati Farimani et al.[24], which model a similar MoS₂ nanopore system.

The identification accuracy depends on the interaction between the type of amino acid and the nanopore. For identifying amino acids with clear structural differences, our accuracy of identification exceeds 90%, such as distinguishing H from D, E (90.62%) and distinguishing P from G, C (92.66%) (Fig. 2b, h). Note that for amino acids with very similar structures (F and Y), our approach can still reach 85.63% (Supplementary Fig. 26a). The accuracy can be potentially improved by optimizing the nanopore geometry because the size and the thickness of nanopore influence the sensitivity, and by reducing the nanopore noise[28]. In addition, the accuracy can be computationally improved by increasing the training dataset.

To prove that the majority of natural amino acids can be discriminated in MoS₂ nanopores, the experiments of 16 amino acids were carried out in the same nanopore (Supplementary Fig. 2). In the buffer with pH = 7.8, K and R carry opposite charge to the other 18 amino acids (Supplementary Fig. 21). We thus excluded K and R for making the comparison, that is, up to 18 amino acids can be compared simultaneously. We managed to compare the signals of 16 out of 18 amino acids in the same nanopore (Device #15). Moreover, the current MoS₂ nanopore system is able to identify one specific amino acid within a mixture (Supplementary Fig. 30).

## The resolution of MoS₂ nanopores for identifying chemical groups

The ultimate resolution of nanopores for the detection of peptides has been continuously updated in recent years. Discrimination among peptides differed by one amino acid substitution has been demonstrated using the engineered FraC nanopores[29], in which the resolution reached 44 Da. The distinction of amino acids carried in a polycationic polymer in aerolysin nanopores has shown remarkable sensitivity[19]. However, these measurements distinguished single amino acids in indirect ways that measure the changes by substitutions. To further explore the ultimate resolution of the current MoS₂ nanopore system, we carried out three groups of experiments for discriminating amino acids with sub-1 Da mass difference (Fig. 3). For D and N, or E and Q, the mass difference between the pairs in each group is only 0.99 Da, and the molecular skeleton is the same except for the end chemical group, where the electrically charged amino acids (D, E) have -OH (17.01 Da), polar uncharged amino acids (N, Q) have -NH₂ (16.02 Da). Though each pair of amino acids has a very close molecular weight or volume, the difference in their functional groups creates a difference in their molecular configuration and carried charges, which leads to distinguishable current blockades in MoS₂ nanopores (Fig. 3a, b, d, e, Supplementary Fig. 31a, b). For L and N with a molecular weight difference of 0.94 Da, their structures differ a lot, and their relative current blockades can also be distinguished (Fig. 3c, f, Supplementary Fig. 31c). To the best of our knowledge, this is the highest resolution in nanopores (discriminating sub-1 Da molecular weight difference) that has been reported experimentally.

## Discriminating amino acid phosphorylation

In principle, sub-1 Da nanopore resolution can provide sufficient spatial resolution for protein sequencing with an ultra-confined nanopore sensing region. However, de novo protein sequencing still requires a way to precisely control the stepping of the peptide through the pore. The introduction of enzymes[17] to biological nanopores for controlling the motion of peptides brings enlightenment, considering the compatibility of DNA polymerases with solid-state nanopore systems is shown feasible[30]. Thanks to the single amino acid discrimination with sub-1 Da resolution in the MoS₂ nanopore system, we can apply the current methodology to identify post-translational modifications of amino acids. The phosphorylation of tyrosine (Y) plays a key regulatory role in cell activity, and the abnormal Y phosphorylation is closely related to cancerization[31]. We analyzed tyrosine (Y) and phosphorylated tyrosine (p-Y) using a MoS₂ nanopore (Fig. 4a–c, Supplementary Fig. 32). As shown in Fig. 4b, the mean values of $\Delta I/I_0$ for Y and p-Y are 0.106 ± 0.012, 0.128 ± 0.019, and the identification accuracy is 80.67%. This demonstration indicates that MoS₂ nanopores have the

potential to recognize phosphorylated amino acids and should not be only limited to detecting phosphorylation but also generally apply to detecting any functional group change of a single amino acid.

## Discussion

To summarize, we have shown that a sub-nanometer engineered MoS₂ nanopore can be used to directly identify single amino acids and recognize the chemical modifications. When the physical dimension of the nanopore system is rationally designed, the nanopore resolution could be greatly improved to a level of sub-1 Dalton. With our MoS₂ nanopore, 16 out of 20 types of natural amino acids can be identified due to the present discrimination capability of MoS₂ nanopores, which could potentially be further improved by optimizing the pore structure, as well as the fact that two types of amino acids (K and R) are positively charged in the current voltage driven experimental system in contrast to the other 18 amino acids, which can be potentially addressed by further implementation of the enzyme-based peptide driven approach[15, 17]. The advance in nanopore resolution, in a future combination with such precise peptide stepping control methods, may pave the way to single-molecule peptide sequencing. Finally, we believe this understanding should not only limit to the MoS₂ pores explored in this work but also illuminate that the atomic engineering of chemically modified biological nanopores, ultrathin solid-state nanopores, or de novo designed molecular nanopores[32] with sub-nanometer sensing region is required for promoting protein sequencing and ultrasensitive chemical analysis with nanopores.

## Methods

### Nanopore fabrication

The procedure for the fabrication of MoS₂ nanopore devices refers to a previously published method[33–35]. In brief, single-layer MoS₂ films grown by chemical vapor deposition were transferred from SiO₂/Si substrates and suspended on micro/nano fabricated SiN$_x$ membranes with a supporting hole of 40–80 nm. Using the electrochemical etching method[34], MoS₂ nanopores were drilled under 0.8–1 V voltages. Flow cells were assembled by two polymethylmethacrylate chambers which sealed the chips with nanopores using two rubber O-rings. H₂O:ethanol solution (1:1, vol:vol) was injected into each chamber, and the nanopore chips were wetted for at least 30 min before performing ionic current measurements. A pair of Ag/AgCl electrodes connected to patch clamp amplifier was used to apply voltages and measure ionic current. An external voltage is applied on the trans side of the chamber, and the cis side is electrically grounded[23]. The bias direction of the voltage depends on the charged nature of the amino acids in the buffer solution (Supplementary Fig. 21).

### Nanopore measurement

An Axopatch 200B patch clamp amplifier (Molecular Devices, USA) was used to record the ionic current. An NI PXI-4461 card was used for data digitalization and data acquisition. Data recording was filtered through a 10-kHz low-pass Bessel filter, and the sampling frequency was fixed to 100 kHz.

### Amino acids identification

1 M KCl solution buffered with 10 mM Tris-HCl and 1 mM EDTA at pH 7.8 was used as the buffer solution unless otherwise specified. Amino acids (Sangon Biotech Co., Ltd., Shanghai, China) were dissolved in the buffer solution and diluted to 2 μM for nanopore experiments. Before each experiment, the two chambers of devices were flushed at least three times to ensure the absence of the analyte residuals from the previous round of the experiment.

### Data analysis

Experimental data were analyzed using Igor Pro 6.12 software (WaveMetrics) and MATLAB R2019a software (MathWorks). The current traces displayed in the figures were downsampled to 10 kHz. The trace in Fig. 1b was processed with a window-based FIR filter (filter order: 1000, frequency constraints: 2000, type: highpass). Event detection was performed using an open-source Matlab code package, Transalyzer[36]. For the parameter setting, we chose the Butter 2$^{nd}$ type with 10 kHz for the filter, and the baseline was calculated by a moving average window of 300 ms. Each type of single amino acid was translocated in at least two different devices, and representative and reproduced results are presented in Supplementary Figs. 22, 24–27. The function in Distribution Fitting Tool (Matlab) was used for generating histograms of relative current blockades and fitting them into curves (Display type: Density, Distribution fit: Normal Kernel).

### Deep neural network

To classify different kinds of amino acids, a deep learning network, called Single Amino Acids Identification Network (SAAINet, Supplementary Software provided), with long short-term memory (LSTM)[37] units, is developed to model the sequential information effectively. LSTM is a special kind of neural network, which is improved by a recurrent neural network (RNN) to mitigate the long learning dependencies problems. With the ability to efficiently capture long-term information with different sequence lengths, deep neural networks with LSTM are widely used in natural language process (NLP)[38], speech recognition[39], and action recognition[40]. The overall structure of our SAAINet model is shown in Supplementary Fig. 16, which consists of the LSTM, the pooling layer, four fully connected layers, and the output layer (refer to Supplementary Note 4 for more details).

### Statistics and reproducibility

Representative reproduced results of single amino acids identification experiments are provided in Supplementary Materials. At least 300 data points were collected for each experimental measurement, and the majority of results include more than 1000 data points. P-values of the z-test on all the data are less than 0.0001. Data was filtered through a 10-kHz filter, and the events were excluded when the dwell time was less than 0.1 ms. All the experiments were performed independently under comparable experimental conditions. No randomization or blinding was used.

### Reporting summary

Further information on research design is available in the Nature Portfolio Reporting Summary linked to this article.

## Data availability

The main data generated in this study are available within the Article, the Supplementary Information file, and the Source Data file. Source data are provided with this paper.

## Code availability

The codes of the Single Amino Acids Identification Network (SAAINet) are available within the Supplementary Software.

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

## Acknowledgements

This work is supported by the National Natural Science Foundation of China (21974123), the National Key R&D Program of China (2020YFA0211200), the Natural Science Foundation of Zhejiang Province (LR20B050002), and the Fundamental Research Funds for the Central Universities (K20220088) to J.F. We thank the Micro and Nano Fabrication Center, the Chemistry Instrumentation Center and the State Key Laboratory of Modern Optical Instrumentation at Zhejiang University for facility support.

## Author contributions

F.W. fabricated the devices, performed the experiments, analyzed the data, and wrote the manuscript; C.Z. fabricated the devices and performed the phosphorylation experiments; P.Z. developed the SAAINet model; F.C. prepared the silicon nitride samples; D.Q. performed data analysis; J.F. conceived the project, supervised the study, and wrote the manuscript.

## Competing interests

J.F. and F.W. have submitted a patent application (application number CN202210909213.0) and a PCT patent application (application number PCT/CN2023/088193) to China National Intellectual Property Administration pertaining to the method for identifying single amino acids and discriminating amino acid phosphorylation in $MoS_2$ nanopores of this work. The remaining authors declare no competing interests.
