## [Peer Review File · Nature Communications]

Editorial Note: This manuscript has been previously reviewed at another journal that is not operating a transparent peer review scheme. This document only contains reviewer comments and rebuttal letters for versions considered at *Nature Communications* .

Reviewer #1 (Remarks to the Author):

The manuscript has been transferred from Nature Nanotechnology to Nature Communications. I had been reviewer of the manuscript submitted to Nature Nanotechnology (reviewer 1) and have read the authors reply to my original set of comments.

The authors have addressed most points.

To become publishable in Nature Communications. the authors should include the reply to points 1.3. and 1.4. as notes in the SI of the manuscript (after editing for conciseness and to remove the references to the reviewer comments).

In addition, the different device characteristics should be listed in a table.

Reviewer #2 (Remarks to the Author):

I am satisfied with the revision and I am happy to recommend the paper for publication in Nature Communications.

Response to Reviewer Comments:

Reviewer #1:

The manuscript has been transferred from Nature Nanotechnology to Nature Communications. I had been reviewer of the manuscript submitted to Nature Nanotechnology (reviewer 1) and have read the authors reply to my original set of comments. The authors have addressed most points.

1.1 To become publishable in Nature Communications, the authors should include the reply to points 1.3. and 1.4. as notes in the SI of the manuscript (after editing for conciseness and to remove the references to the reviewer comments).

We thank the reviewer for the suggestions. We have added our response to points 1.3. and 1.4. as notes in the SI of the manuscript (see Supplementary Note 2 “Pore size and region of sensitivity”).

1.2 In addition, the different device characteristics should be listed in a table.

The different device characteristics have been displayed in SI figures. 8-9, and we have numbered each devices in the relevant figure/figure captions.

Reviewer #2:

I am satisfied with the revision and I am happy to recommend the paper for publication in Nature Communications.

We thank the reviewer for the recommendation.